# Oral Microbiota from Periodontitis Promote Oral Squamous Cell Carcinoma Development via $\gamma\delta$ T Cell Activation

Wei Wei,[a] Jia Li,[a] Xin Shen,[a] ©Jinglu Lyu,[a] Caixia Yan,[a] Boyu Tang,[a] Wenjuan Ma,[a] Huixu Xie,[a] Lei Zhao,[a] Lei Cheng,[a] ©Ye Deng,[b] Yan Li[a]

[a]State Key Laboratory of Oral Diseases, National Clinical Research Center for Oral Diseases, West China Hospital of Stomatology, Sichuan University, Chengdu, China
[b]CAS Key Laboratory of Environmental Biotechnology, Research Center for Eco-Environmental Sciences, Chinese Academy of Sciences, Beijing, China

**ABSTRACT** Oral squamous cell carcinoma (OSCC) is a fatal disease, and periodontitis is associated with OSCC development. However, the pathogenesis in the context of OSCC with periodontitis has not been fully understood. Here, we demonstrated that periodontitis promoted OSCC development, accompanied by alterations in the oral bacterial community and the tumor immune microenvironment. The oral microbiota from periodontitis maintained the dominant position throughout the whole process of OSCC with periodontitis, of which *Porphyromonas* was the most abundant genus. The oral microbiota from periodontitis could activate interleukin-17-positive (IL-17$^+$) $\gamma\delta$ T cells directly. The activated $\gamma\delta$ T cells were necessary for the IL-17/signal transducer and activator of transcription 3 (STAT3) pathway and promoted M2-tumor-associated macrophage (TAM) infiltration in OSCC proliferation. Our data provide insight into the carcinogenesis of OSCC with periodontitis by outlining the tumor-associated immune response shaped by the oral microbiota from periodontitis. Thus, oral commensal bacteria and IL-17$^+$ $\gamma\delta$ T cells might be potential targets for monitoring and treating OSCC.

**IMPORTANCE** The work reveals the role of the oral microbiota from periodontitis in carcinogenesis. Furthermore, our study provides insight into the pathogenesis of OSCC with periodontitis by outlining the tumor-associated immune response shaped by the oral microbiota from periodontitis, which might identify new research and intervention targets for OSCC with periodontitis.

**KEYWORDS** periodontitis, microbiota, OSCC, $\gamma\delta$ T, IL-17, STAT3

Oral cancer is one of the most damaging malignant diseases worldwide, with a high incidence and recurrence rate (1). Most oral cancer cases are oral squamous cell carcinoma (OSCC). With the recognition that microbial pathogens contribute to cancer, the role of bacteria in OSCC has attracted research interest. Periodontitis is an important chronic oral disease that can cause illness in nearby and distant tissues and organs. There are dysbiotic transitions in the oral commensal communities in periodontitis (2), including subgingival plaque, saliva, tongue dorsum, buccal mucosa, etc. (2–5). Dozens of studies have shown that periodontitis and oral cancer are related (6, 7).

Mucosal compartments such as the oral cavity, trachea, and intestines are colonized by a vast number of microbes in the external environment, which can influence immunotherapeutic interventions for cancers (8, 9). The mainstream academic view holds that the mechanisms by which bacteria promote carcinogenesis in OSCC with periodontitis may include the following (10): (i) carcinogen production, (ii) apoptosis inhibition, (iii) cell proliferation activation, (iv) cell invasiveness enhancement, and (v) chronic inflammation induction. Several studies have confirmed the existence of links between OSCC and some specific bacterial species, such as *Porphyromonas gingivalis* and *Fusobacterium nucleatum* (11–14). However, knowledge regarding the changes in the oral microbial community and its role in the context of OSCC with periodontitis is

**Ad Hoc Peer Reviewer** Vivek Thumbigere-Math,, University of Maryland School of Dentistry

Address correspondence to Yan Li, feifeiliyan@163.com.

The authors declare no conflict of interest.

comparatively lacking. Robust, reproducible, and putative mechanisms of oral microbiota from periodontitis in carcinogenesis have not been identified.

Microorganism disorders can distort the network of immune cells, make it lose the function of stabilizing mucosal tissue, and even participate in tumors. As a key subset in oral barrier immunosurveillance, γδ T cells can recognize multiple pathogen antigens (15). However, the effect of γδ T cells is controversial because of their different subtypes (16). Interleukin-17-positive (IL-17$^+$) γδ T cells are the primary source of interleukin-17 (IL-17), which has immunosuppressive effects and promotes cancer progression directly (17–20). Some specific subsets of γδ T cells also have antitumor activity due to their cytotoxic capacity and gamma interferon (IFN-γ) production (21).

The mechanism underlying the development of OSCC may involve the interactive response of three factors: oral microbiota, the immune system, and cancer tissues. The mechanism by which the oral microbiota interfere with the tumor immune response to regulate the microenvironment in OSCC with periodontitis has not been fully understood. Here, we focused on identifying the characteristics of the oral microbial community in the context of OSCC with periodontitis and exploring the mechanistic regulatory axis of γδ T cells in the context of OSCC with periodontitis. Our data provide novel insight into the pathogenesis of OSCC with periodontitis by outlining the tumor-associated immune response shaped by the oral microbiota from periodontitis. This may identify new research and intervention targets for OSCC with periodontitis.

## RESULTS

**Experimental periodontitis treatment promotes the development of OSCC.** To address whether periodontitis plays a protumor role in OSCC, we established a mouse model of OSCC with ligature and infection by oral microbes from periodontitis (22). The group with periodontitis (P or OP group) displayed significantly increased alveolar bone resorption compared to the groups without periodontitis (C or O group) (see Fig. S1a in the supplemental material). Seven days after SCC7 cell inoculation into the buccal mucosa, we found OSCC in the mouths of the mice. Compared with mice in the OSCC group (O group), mice in the OSCC and periodontitis group (OP group) exhibited increased tumor weights and cell proliferation, as demonstrated by Ki67 staining (Fig. 1a).

To determine the functional importance of periodontitis in OSCC progression, we compared tumor development among OSCC mice with periodontitis or not at different stages (Fig. 1b). At the early stage of OSCC (7 days after SCC7 cell inoculation), mice with OSCC and periodontitis (EOP group) showed higher tumor weights, higher tumor volumes, and higher tumor cell proliferation levels than mice with only OSCC (EO group) (Fig. 1b). At advanced stages, treatment with periodontitis (AOP group) robustly promoted tumor growth, with increased tumor weights, increased expression of Ki67, and decreased survival rates (Fig. 1b and Fig. S1b).

In contrast with treating mice with foreign, periodontitis-related, and pathogenic microorganisms, antibiotic treatment did not affect tumor growth and even played a role in remission in many cancers. Astonishingly, our data showed that the antibiotic cocktail (4Abx) was correlated with the OSCC tumor burden significantly (Fig. 1b and Fig. S1b). Furthermore, we found that ligation alone did not affect tumor size and proliferation level without the inoculation of oral microbes from periodontitis (Fig. S1c). The above results emphasized the correlation between affecting the oral microbial community and the development of OSCC, which drew our attention to the role of oral microbiota in the context of OSCC. These findings demonstrated that experimental periodontitis treatment played a profound role in stabilizing mucosal tissues and suggested that the oral microbiota might have a connection with the development of OSCC.

**The oral bacterial community exhibits a persistent and distinct alteration in OSCC with periodontitis.** To ascertain the link between the oral microbiota and OSCC development, 16S rRNA sequencing was performed on the saliva of mice with OSCC, including mice in the EO, EOP, EOA, AO, AOP, and AOA groups.

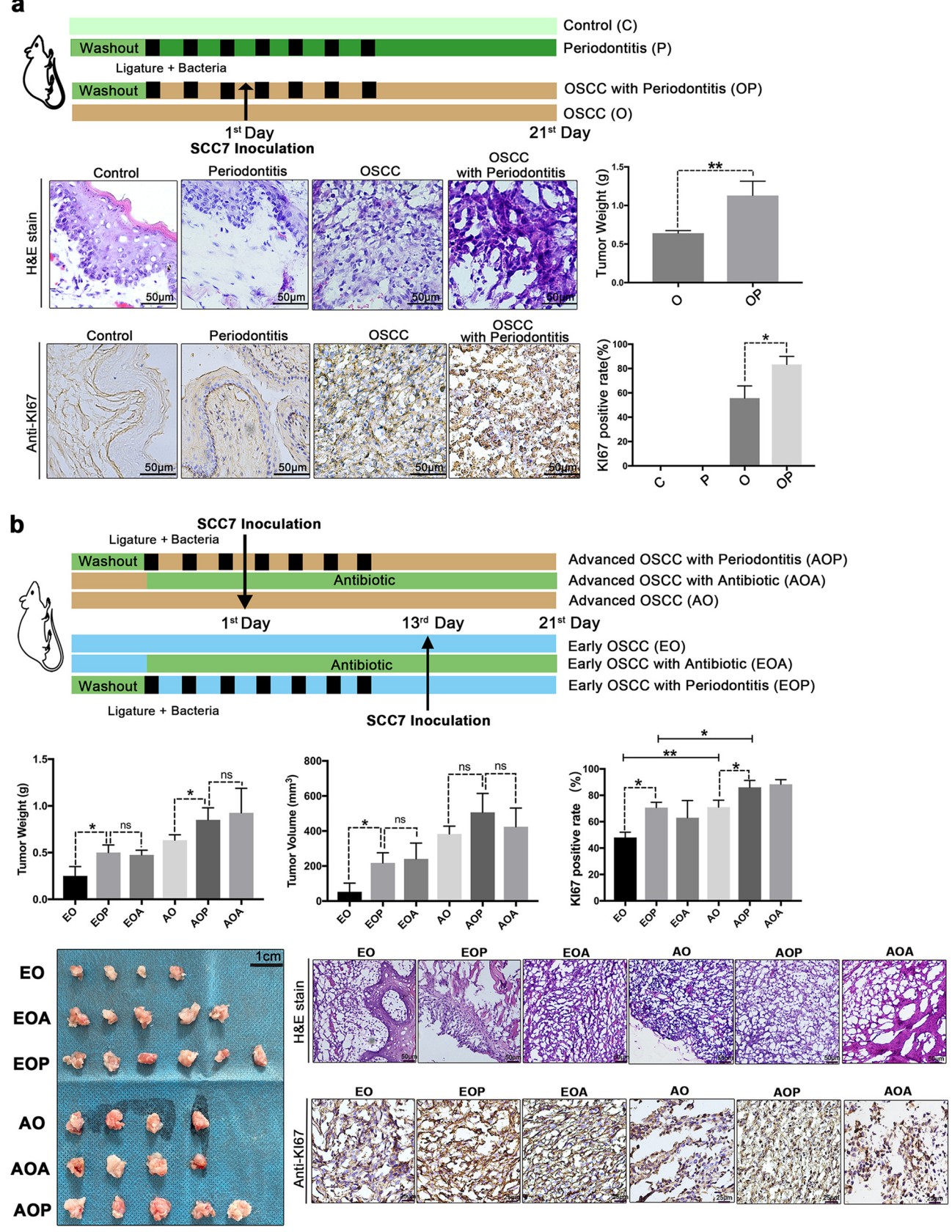

FIG 1 Experimental periodontitis treatment promotes the development of OSCC. (a) Specific-pathogen-free (SPF) mice were subjected to ligature and infection with oral microbes from periodontitis patients. Histopathological analysis and Ki67 staining were performed to examine tumor cell

Compared to the early stage of OSCC (EO group), the advanced stage (AO group) exhibited reduced $\alpha$ diversity in the oral commensal bacterial community, as measured by the Chao index, observed richness, evenness, and phylogenetic diversity (Fig. 2a). As expected, the $\alpha$ diversity of the oral microbiome decreased under the 4Abx treatment (EOA and AOA group) (Fig. 2a). Of particular importance, at the early stage of OSCC, mice with OSCC and periodontitis (EOP group) displayed a significantly decreased $\alpha$ diversity of the oral microbiome compared to that of their counterparts without periodontitis (EO group), although there was no significant difference in $\alpha$ diversity between the early (EOP group) and the advanced (AOP group) stage of OSCC with periodontitis (Fig. 2a).

Analysis of the $\beta$ diversity by principal-coordinate analysis (PCoA) showed a different distribution between the EO group and the AO group, which implied the oral microbiome could be changed due to OSCC development (Fig. 2b). The similarity in bacterial profiles between the EOP and AOP groups demonstrated that the oral microbiota from periodontitis effectively and persistently separated the bacterial distribution from that of the group without periodontitis (Fig. 2b). In terms of altering the bacterial community, the oral microbiota from periodontitis were more capable than OSCC development.

Sequencing data showed the composition of each group was different. Moreover, some particular oral microbes from periodontitis could be the dominant "king" in the entire process of tumor development and were not easily affected. Compared to the groups with OSCC alone (EO and AO groups), the phylum *Bacteroidota* was the most abundant bacterial phylum in both the EOP group (92.88%) and the AOP group (48.33%) (Fig. 2c). In the early stage of OSCC (EO group), *Streptococcus*, *Staphylococcus*, and *Corynebacterium* were significantly overrepresented (Fig. S2a). On the other hand, *Porphyromonas* was the most abundant genus in the EOP (92.46%) and AOP (47.58%) groups (Fig. 2c). *Fusobacterium*, *Neisseria*, and *Leptotrichia* were more abundant in the AOP group (Fig. S2b). *Citrobacter* and *Streptococcus* were the most prevalent genera in the EOA and AOA groups, respectively (Fig. 2c). Additionally, we performed gene enrichment analysis on samples with sequencing data for bacterial colony markers with the KEGG pathway database (Fig. S2c).

These findings establish a link between the development of OSCC and oral microbiota dysbiosis. The introduction of the oral microbiota from periodontitis dramatically altered the oral commensal bacterial community in OSCC and might accelerate disease development continuously as a relatively stable role.

**In the development of OSCC with periodontitis, IL-17$^+$ $\gamma\delta$ T cells are activated.** To further investigate the immune mechanism in the progression of OSCC with periodontitis, we analyzed the immune cytokines in the tumor tissues of each group. The reverse transcription-quantitative PCR (qRT-PCR) results showed that numerous genes, including *Ifn$\gamma$*, *Il6*, and *Il1$\beta$*, were downregulated in the EOP or the AOP group, compared to the EO or the AO group, respectively (Fig. 3a). The expression of IFN-$\gamma$ in serum was nearly abolished in the AO group (Fig. 3b). Notably, the expression level of IL-17A was increased in advanced OSCC (AO group) and had a further significant increase in the AOP group (Fig. 3a and b). Moreover, gene expression analyses using the TCGA database showed that IL-17RA mRNA levels were significantly higher in head and neck squamous cell carcinoma (HNSC) tissues than in the corresponding normal tissues (Fig. S3a).

$\gamma\delta$ T cells are a major source of IL-17 (19, 23). IL-17A expression was significantly correlated with the infiltration levels of $\gamma\delta$ T cells in HNSC (Fig. S3a). In our experiment, the changes in $\gamma\delta$ T cells were striking. When OSCC progressed to an advanced stage, the

**FIG 1** Legend (Continued)
proliferation. Representative pictures and corresponding quantitative data are shown. (b) SPF mice were subjected to ligature and infection with oral microbes from periodontitis or treated with 4Abx in the drinking water before tumors developed. OSCC tissues were analyzed on the 7th day or 21st day after injection. Representative pictures and corresponding quantitative data are shown. C, control group; P, periodontitis group; O, SCC7 cell inoculation-treated group; OP, periodontitis and SCC7 cell inoculation-treated group; AO, 21 days after tumor initiation; AOP, the AO group treated with periodontitis; AOA, the AO group treated with 4Abx; EO, 7 days after tumor initiation; EOP, the EO group treated with periodontitis; EOA, the EO group treated with 4Abx. The results are expressed as the mean ± SD. ns, not significant; *, $P < 0.05$; and **, $P < 0.01$, by ANOVA. The data represent the results of over 3 independent biological replicates. Number of mice in each group: C, $n = 5$; P, $n = 5$; O, $n = 5$; OP, $n = 5$; EO, $n = 5$; EOA, $n = 5$; EOP, $n = 6$; AO, $n = 6$; AOA, $n = 6$; AOP, $n = 8$. See also Fig. S1.

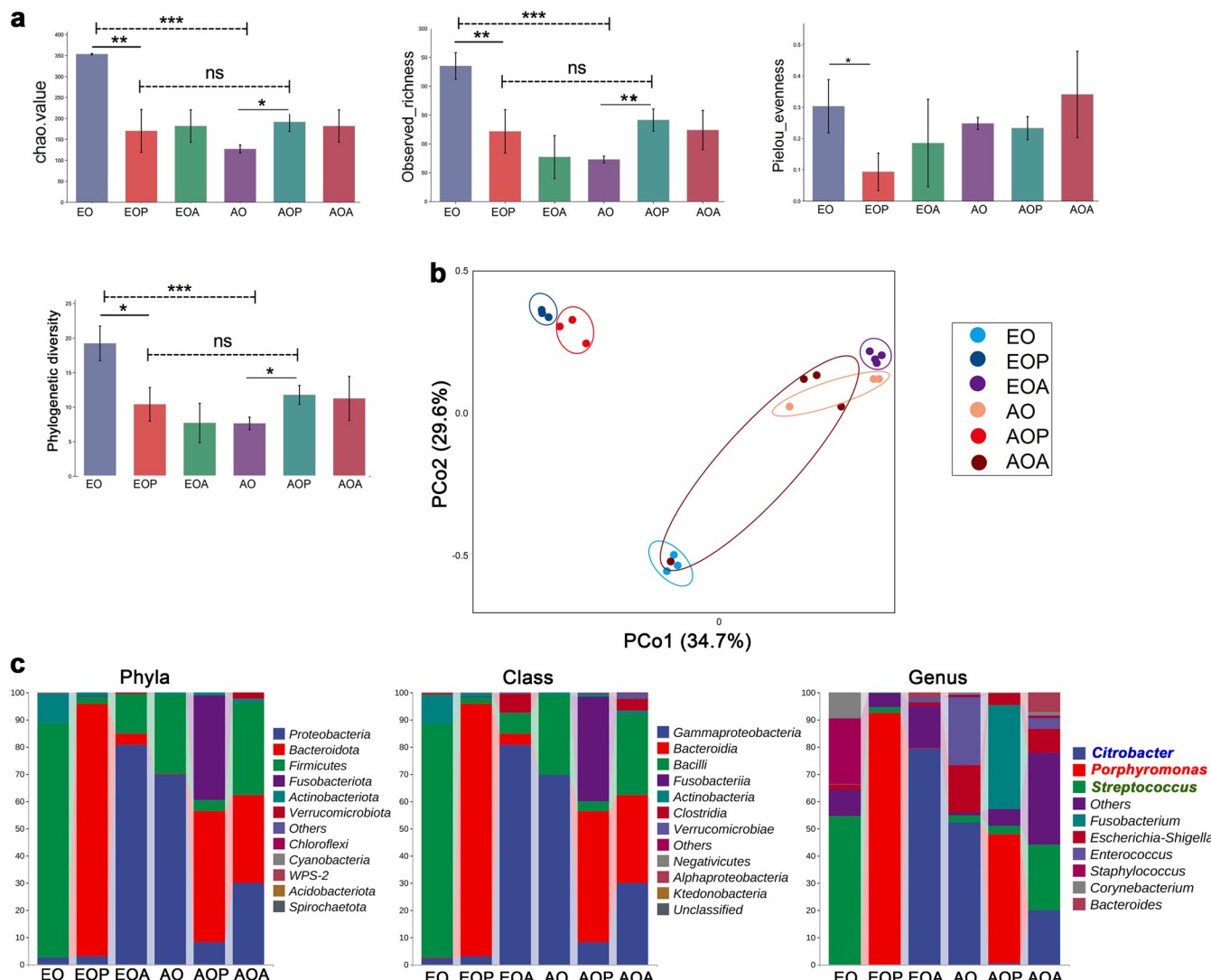

**FIG 2** The oral bacterial community exhibits a persistent and distinct alteration in OSCC with periodontitis. (a and b) The diversity and structure of the oral microbiota in each group. The α diversity (Chao index, richness, evenness, and phylogenetic diversity) of the oral microbiota in each group. ns, not significant; *, $P < 0.05$; **, $P < 0.01$; and ***, $P < 0.001$, by the Kruskal-Wallis test and Dunn's test. The β diversity is shown as a PCoA plot of UniFrac distances. (c) The percentages of the major phyla, classes, and genera represented by the oral microbiota, as determined by 16S rRNA sequencing. AO, 21 days after tumor initiation; AOP, the AO group treated with periodontitis; AOA, the AO group treated with 4Abx; EO, 7 days after tumor initiation; EOP, the EO group treated with periodontitis; EOA, the EO group treated with 4Abx. Each group had 3 to 4 individual samples analyzed by 16S rRNA sequencing. See also Fig. S2.

proportion of IL-17⁺ γδ T cells increased significantly, accounting for 20% to 40% of the total γδ T cell population in the tumor tissues. Importantly, more γδ T cells produced the cytokine IL-17 in both the early stage (EOP group) and the advanced stage (AOP group) of OSCC with periodontitis (Fig. 3c). Immunofluorescence analysis again confirmed the above results (Fig. 3c). The significant relationships between immune factors and the microbial community were assessed by canonical correlation analysis (CCA) (Fig. S3c). The CCA results indicated that tumor volume ($P = 0.021$), γδ T ($P = 0.016$), IL-17A ($P = 0.044$), IFN-γ ($P = 0.033$), and TLR4 ($P = 0.01$) were significant factors for the microbial community, implying that these factors are associated with the oral microbial community.

Consistent with Fig. S1c, the obvious activation of IL-17⁺ γδ T cells was not observed without the inoculation of oral microbes from periodontitis (Fig. S3d). Collectively, these data demonstrated that in the development of OSCC with periodontitis, IL-17⁺ γδ T cells were activated. The oral microbiota from periodontitis and IL-17⁺ γδ T cells might be closely related.

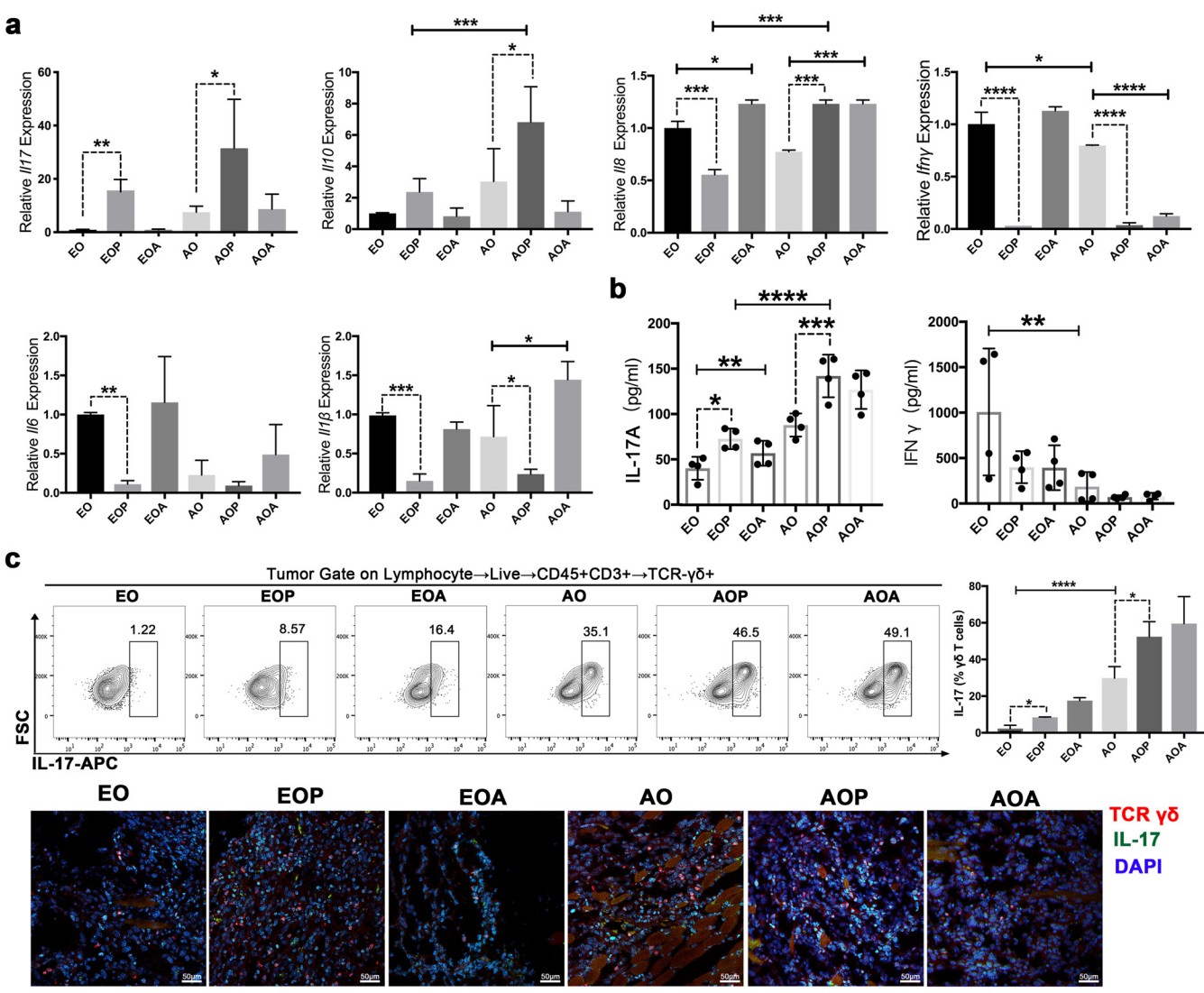

**FIG 3** In the development of OSCC with periodontitis, IL-17⁺ γδ T cells are activated. (a) The mRNA expression of *Il17, Il10, Il18, Ifnγ, Il6,* and *Il1β* in the tumor tissues was detected via qRT-PCR, and expression was normalized to *Gapdh*. (b) IL-17A and IFN-γ levels in serum were measured by ELISA. (c) IL-17 expression in γδ T cells from OSCC tissues was examined by flow cytometry. Representative plots are shown, and the frequencies of IL-17⁺ γδ T cells were quantified. Representative confocal immunofluorescence images of OSCC tissues are shown. Red, TCR γδ; green, IL-17. AO, 21 days after tumor initiation; AOP, the AO group treated with periodontitis; AOA, the AO group treated with 4Abx; EO, 7 days after tumor initiation; EOP, the EO group treated with periodontitis; EOA, the EO group treated with 4Abx; FSC, forward scatter; APC, allophycocyanin. The results are expressed as mean ± SD. ns, not significant; *, $P < 0.05$; **, $P < 0.01$; ***, $P < 0.001$; and ****, $P < 0.0001$, by ANOVA. The data represent the results of over 3 independent biological replicates. See also Fig. S3.

**Oral microbiota from periodontitis activate γδ T cells to produce IL-17 and stimulate cell proliferation.** To evaluate the responses of immune cells to the oral microbiota from periodontitis, total peripheral blood mononuclear cells (PBMCs) from mice were incubated with oral microbiota from periodontitis (Fig. 4a). Flow cytometry was used to identify γδ T cells and IL-17⁺ γδ T cells. After incubation with the live oral microbiota from periodontitis for 6 h, both γδ T and IL-17⁺ γδ T cells proliferated more than the counterpart cells incubated without stimulators (Fig. 4a).

Because of the various virulence compositions derived from oral microbiota from periodontitis, we examined the effects of multiple bacterial components on the activation of IL-17⁺ γδ T cells separately. As shown in Fig. 4b, even with a filter physically separating the bacteria, the increased proportion of IL-17⁺ γδ T cells was induced by live oral microbiota from periodontitis. However, the same concentrations of ultrasonicated or heat-inactivated microbiota had no significant effect on the activation of IL-17⁺ γδ T cells, suggesting the involvement of factors secreted by live microbiota. Because IL-17 is an important cytokine

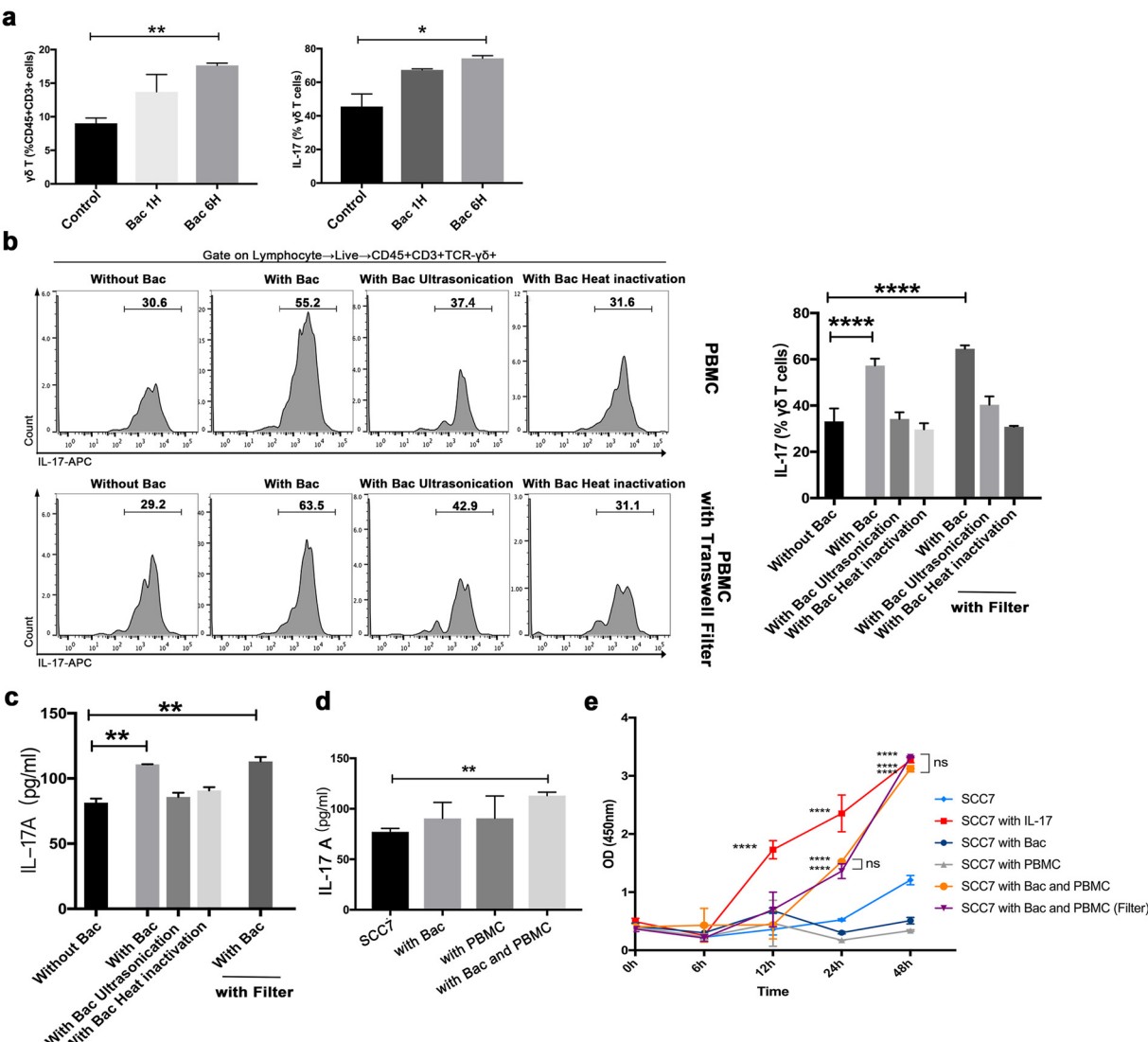

**FIG 4** Oral microbiota from periodontitis activate γδ T cells to produce IL-17 and stimulate cell proliferation. (a) PBMCs were cocultured with live oral microbiota from periodontitis for 1 or 6 h. The frequencies of γδ T and IL-17⁺ γδ T cells were quantified. (b) With or without filters, PBMCs were cocultured with the microbiota treated in different ways. The frequencies of IL-17⁺ γδ T cells were quantified, and the representative histograms of flow cytometry are shown. (c) IL-17A levels of medium supernatant in panel b were measured. (d) SCC7 cells were cocultured with PBMCs and the live oral microbiota from periodontitis. IL-17A levels in the culture supernatant were measured. (e) CCK-8 assays were conducted to detect the cell viability of SCC7 cells. Results are expressed as the mean ± SD. *, $P < 0.05$; **, $P < 0.01$; ***, $P < 0.001$; and ****, $P < 0.0001$, by ANOVA; data represent at least 3 independent experiments. See also Fig. S4.

produced by γδ T cells, we examined the concentration of IL-17A in the supernatant under each culture condition. A significant increase in IL-17A was observed after coculture with the live oral microbiota from periodontitis (Fig. 4c). To further demonstrate the effect of bacterial infection on IL-17A production in the tumor environment, PBMCs were cocultured with oral microbiota from periodontitis and SCC7 cells for 6 h. IL-17A protein levels in the culture supernatant were measured by enzyme-linked immunosorbent assay (ELISA). As predicted, the concentration of IL-17A was highest in the coculture of immune cells, bacteria, and cancer cells (Fig. 4d). At the same time, we looked into the specific effects of *Porphyromonas. gingivalis*. Although the proportion of γδ T cells in PBMCs did not increase after coculturing with *P. gingivalis*, IL-17⁺ γδ T cells proliferated more than those in the group without *P. gingivalis* (Fig. S4a). At the same time, a significant increase in IL-17A was observed (Fig. S4b).

The high level of IL-17-producing cells in the tumor microenvironment is a poor prognostic factor (19, 24). We investigated whether IL-17 increased the proliferation of

SCC7 cells. The vitality of SCC7 cells in the presence of IL-17 was detected by a CCK-8 assay, and the result demonstrated the crucial role of IL-17 in SCC7 cell proliferation (Fig. 4e). When SCC7 cells were cultured with the microbiota and PBMCs, a higher optical density (OD) value was also observed after 24 h, even with a filter (Fig. 4e), which might be attributed to the high level of IL-17. These *in vitro* results suggest that IL-17$^+$ $\gamma\delta$ T cells were activated by the oral microbiota from periodontitis. In addition, IL-17 in response to the oral microbiota from periodontitis could stimulate SCC7 cell proliferation.

**The activated $\gamma\delta$ T cells are required for the development of OSCC with periodontitis.** The above-mentioned *in vivo*, *in vitro*, and *in silico* analysis data indicated that the microbiota, $\gamma\delta$ T lymphocytes, and IL-17 played important functional roles in the development of OSCC with periodontitis. Further supporting the connection between the oral microbiota from periodontitis and $\gamma\delta$ T cells, we ligated the OSCC mice and inoculated them with the oral microbiota from healthy people (AON group). Compared with the AON group, the oral microbiota from periodontitis significantly induced the growth of tumors (Fig. 5a), the expansion of IL-17$^+$ $\gamma\delta$ T cells (Fig. 5b), and the expression of IL-17A (Fig. 5c).

To assess the functional importance of $\gamma\delta$ T cells, mice bearing tumors were administered six consecutive treatments with monoclonal antibodies against $\gamma\delta$ T cells. As verified by flow cytometric and immunofluorescence analyses (Fig. S5), $\gamma\delta$ T cell expansion was successfully suppressed (AOP-Anti group). Mechanistically, inhibition of $\gamma\delta$ T cells substantially reduced tumor development (Fig. 5a) and IL-17A level (Fig. 5c). In addition, in tumor tissue from the AOP-Anti group, the abundance of M2-tumor-associated macrophages (TAMs), which play major roles in tumor initiation, growth, development, and metastasis, was decreased (Fig. S5b). Our results showed that the oral microbiota from periodontitis-induced $\gamma\delta$ T cells were required for IL-17 production, M2-TAM infiltration, and tumor development in periodontitis-promoted OSCC.

**$\gamma\delta$ T cells regulate the IL-17/STAT3 pathway in OSCC with periodontitis.** IL-17 signaling directly activates cell proliferation via the signal transducer and activator of transcription 3 (STAT3) signaling (25–27). In many cancers, STAT3 is activated and plays a pivotal role in cellular proliferation, invasion, migration, and angiogenesis (28). Targeting the STAT3 signaling pathway is a promising therapeutic strategy for numerous cancers (29). Moreover, TCGA database analysis showed that the expression of both IL-17A and IL-17RA in HNSC was significantly correlated with the expression of STAT3, suggesting that IL-17A binds to IL-17RA on tumor cells and activates downstream transcription factors (e.g., STAT3) to promote tumor infiltration and angiogenesis (Fig. 6a).

As STAT3 was the main effector of multiple oncogenic signaling pathways (30), its phosphorylation level (pSTAT3) was detected by immunofluorescence. We first stimulated the SCC7 cells with IL-17 to identify its connection with pSTAT3 in SCC7 cells. Stimulation with IL-17 resulted in significantly increased pSTAT3 production (Fig. 6b). When PBMCs and the microbiota or *P. gingivalis* were cocultured with oral cancer cells, a similar phenomenon of pSTAT3 in these cancer cells was observed (Fig. 6b and Fig. S4c).

To further understand the regulatory mechanism of $\gamma\delta$ T cells on the IL-17/STAT3 pathway, we characterized the expression of STAT3 in OSCC tissues by Western blotting. Our data demonstrated that, with the expansion of IL-17$^+$ $\gamma\delta$ T cells and increased expression of IL-17A, the STAT3 phosphorylation was also activated in both the early stage and late stage of OSCC with periodontitis (Fig. 6c). We further explored the relationship between the pSTAT3 level and the inhibition of $\gamma\delta$ T cells. Notably, under $\gamma\delta$ T cell inhibition conditions, the decreasing trend in the pSTAT3 level was consistent with the changes in the IL-17$^+$ $\gamma\delta$ T cell abundance and IL-17A expression (Fig. 6c).

Moreover, we performed periodontal examinations in OSCC patients and intraoperatively collected the tumor tissues. In the OSCC-with-periodontitis group, both the proportion of IL-17$^+$ $\gamma\delta$ T cells and the phosphorylation of STAT3 were higher (Fig. 6d). These findings demonstrated that the activated $\gamma\delta$ T cells were required to regulate the IL-17/STAT3 pathway in OSCC with periodontitis.

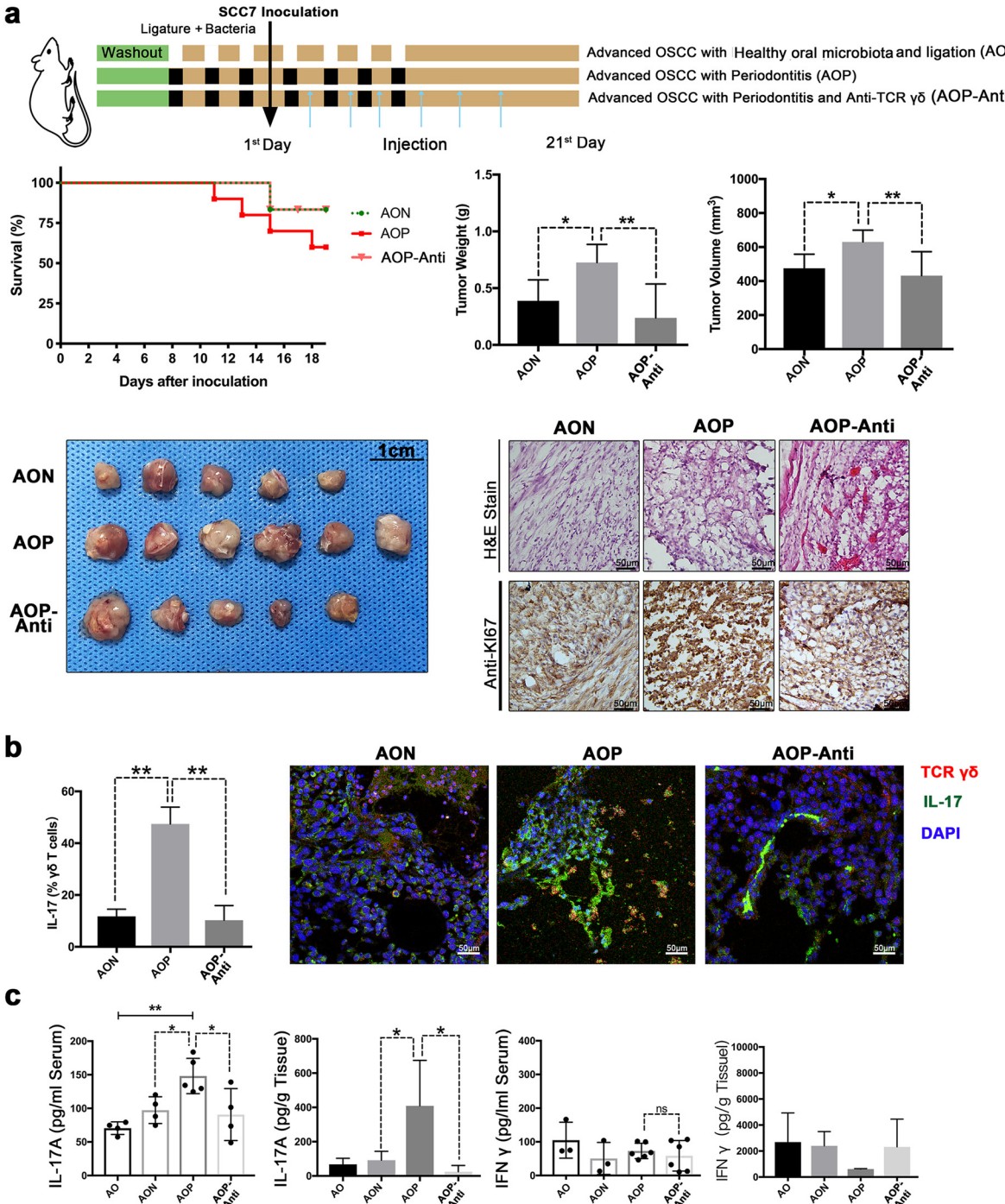

**FIG 5** The activated γδ T cells are required for IL-17A expression and OSCC development. (a) SPF mice were ligated and infected with oral microbiota from periodontitis patients or healthy people. After treatment with the monoclonal antibody UC7-13D5, the OSCC tissues were collected at 21 days after SCC7 cell inoculation. The analysis of survival rate (number of mice in each group: AON, n = 6; AOP, n = 10; AOP-Anti, n = 6), the weight and volume of OSCC tissues, H&E staining, and immunohistochemistry analysis of Ki67 were performed, and representative pictures are shown. (b) IL-17 expression in γδ T cells from OSCC tissues was examined by flow cytometry and immunofluorescence staining. Red, TCR γδ; green, IL-17. (c) IL-17A and IFN-γ levels in serum and OSCC tissues were measured by ELISA. AO, 21 days after tumor initiation; AON, the AO group treated with oral microbiota from healthy people and ligature; AOP, the AO group treated with periodontitis; AOP-Anti, the AOP group with UC7-13D5 injection to inhibit γδ T cells. *, P < 0.05; and **, P < 0.01, by ANOVA. The data represent the results of over 3 independent biological replicates. See also Fig. S5.

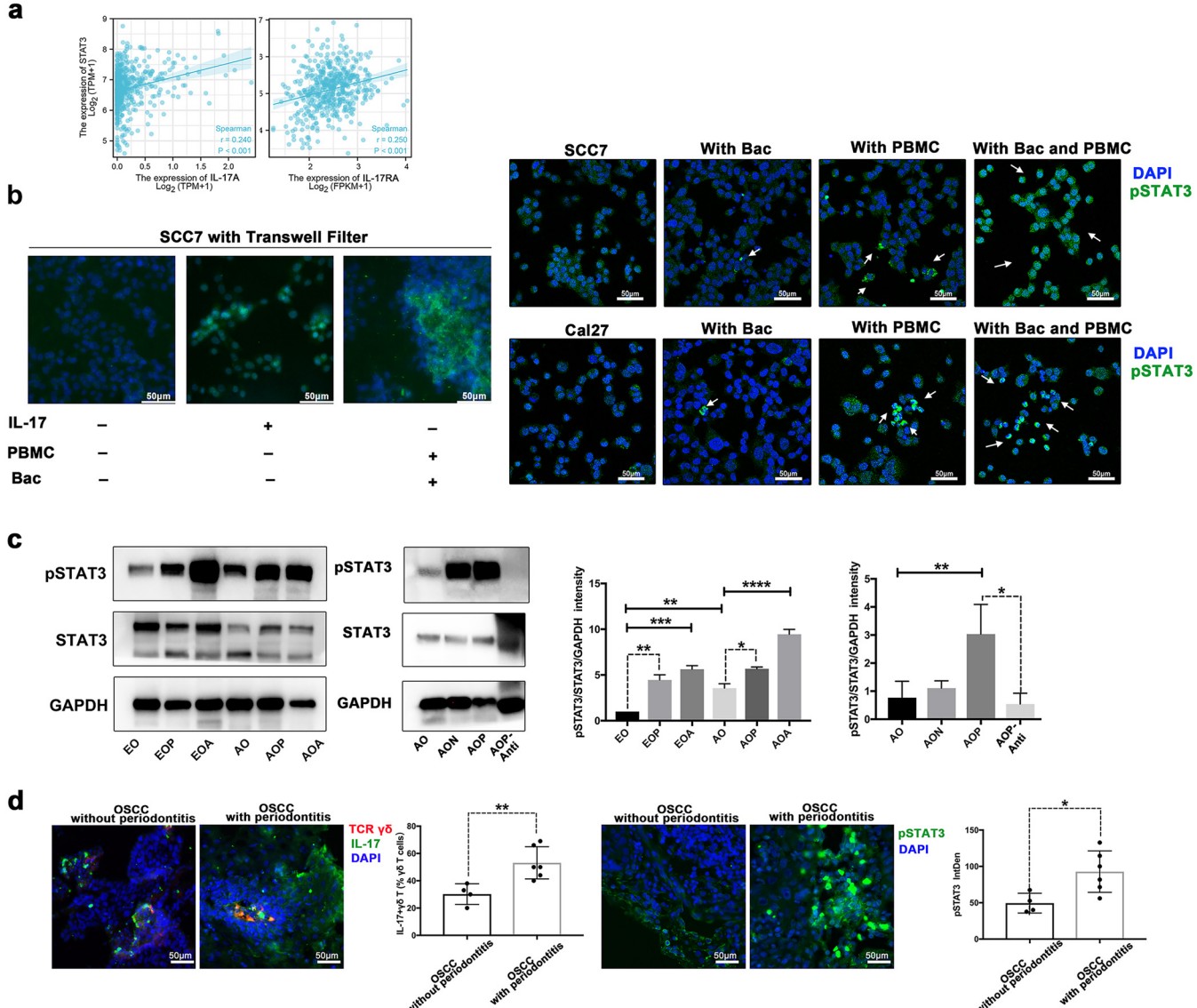

**FIG 6** γδ T cells regulate the IL-17/STAT3 pathway in OSCC with periodontitis. (a) Correlation analysis of IL-17A or IL-17RA and STAT3 gene expression level in HNSC. TPM, transcripts per million; FPKM, fragments per kilobase per million. (b) With or without filters, SCC7 cells or Cal27 cells were cocultured with 200 pg/mL IL-17, the microbiota, and PBMCs (from mice or humans), respectively. The level of pSTAT3 was analyzed by immunofluorescence staining. Green, pSTAT3; the white arrows indicate pSTAT3⁺ cells. (c) Western blot analysis of pSTAT3 and STAT3 in OSCC tissues of different groups. Representative images and a summary of the normalized quantitative data are shown. The results are expressed as the mean ± SD. ns, not significant; *, $P < 0.05$; **, $P < 0.01$; ***, $P < 0.001$; and ****, $P < 0.0001$, by ANOVA. The data represent the results of over 3 independent biological replicates. (d) IL-17 expression in γδ T cells from clinical specimens was analyzed by immunofluorescence staining and quantitated. Red, TCR γδ; green, IL-17. The level of pSTAT3 from clinical specimens was analyzed by immunofluorescence staining and quantitated. Green, pSTAT3. Number of clinical samples: OSCC without periodontitis, $n = 4$; OSCC with periodontitis, $n = 6$. *, $P < 0.05$; and **, $P < 0.01$, by Student's *t* test.

## DISCUSSION

The oral cavity is a mucosal compartment colonized by a complicated microbial community. The microbiota have either pro- or antitumor effects in various tumors, but most of the research has focused on gastrointestinal, pancreatic, cervical, and lung cancers (31). Our data showed the following. (i) During tumor development, the oral microbiota changed constantly. The proportions of *Streptococcus*, *Staphylococcus*, and *Corynebacterium* decreased with the growth of tumors. (ii) Antibiotic treatment reduced the diversity of the oral microbiota, which shaped the oral microbiota and promoted the development of tumors. Oral antibiotic treatment is also associated with an increased risk of colon cancer and immunotherapy failure (32, 33). This is an essential consideration in the clinical treatment of oral cancer patients with antibiotics.

Direct, uncontrolled, and unnecessary interference with the microbiota causes consequences that conflict with the original intent. Overuse of antibiotics is an urgent public health problem, and we have a responsibility to explore better options for the treatment of infections and maintain the integrity of natural microbiome homeostasis. (iii) After inoculation of the oral microbiota from periodontitis, some particular oral microbiome members quickly became prominent members of the oral microbial community. As a continuous disadvantage, the pathogenic structure of the microbiota persisted throughout the process of tumor development, from the early stage to the advanced stage. Furthermore, our experiment indicated, compared with periodontitis oral bacteria, the changes in the mouse native oral microbiota and mouse immunity caused by ligation alone or the healthy human oral bacteria did not significantly contribute to the development of OSCC (Fig. 5 and see Fig. S1c in the supplemental material). Some studies showed that the key human periodontitis oral bacteria, such as *Porphyromonas* and *Fusobacterium*, were associated with the development of human OSCC (34, 35). In conclusion, we thought that although the human oral bacteria invaded the native oral ecology of mice, certain pathogenic oral bacteria of periodontitis did play a key role in promoting OSCC. Based on predicting the composition of functional units, we could get an overview of the functional potential of the samples, and these analysis results might help guide the design of subsequent experiments (such as metagenomic sequencing) (36).

γδ T cells are an important subset of T cells, assisting immune surveillance, tissue repair, and homeostasis (37). In periodontitis, bacteria induce γδ T cells to infiltrate strategically. However, *in vivo* studies have shown that γδ T cells both exert protective effects during age-related bone loss and promote bone resorption in experimental periodontitis (38). In diverse tumors, γδ T cells are the source of the cytokines IL-17 and IFN-γ, which lead to a striking dichotomy of γδ T cells (39, 40). IL-17$^+$ γδ T cells are the primary source of interleukin-17 (IL-17), which has immunosuppressive effects and promotes cancer progression directly (17–20). Some specific subsets of γδ T cells also have antitumor activity due to their cytotoxic capacity and IFN-γ production (21). However, the explanation of the role of γδ T cells in OSCC is unclear. Our study revealed that the presence of the oral microbiota from periodontitis modified the number and function of tissue-resident γδ T cells. As the proportion of IL-17$^+$ γδ T cells increased, the expression level of IL-17, the proportion of M2-TAMs, and the volume of tumors also increased under the nonhomeostatic conditions, but the expression level of IFN-γ did not. When we inhibited γδ T cells, all the above-mentioned effects were reversed, proving that γδ T cells are key to the process by which the oral microbiota from periodontitis promotes OSCC development. The additional mechanistic insights of the interplay between the microbiota and γδ T cells remain to be exploited in the future. Shi et al. discovered a close positive correlation between γδ T cells and the α diversity of the microbiota in the lungs of cancer patients (41, 42). Wilharm et al. found that ablation of γδ T cells alters the relative diversity of oral microbiota in specific-pathogen-free (SPF) B6 mice (43). We will further analyze the microbiota after inhibiting γδ T cells by metagenome and metabolome sequencing in the future.

IL-17 has a proinflammatory function in various chronic diseases (44). Recently, evidence has suggested that IL-17 can stimulate cancer cell migration and invasion in many cancers (45). STAT3 is an oncogenic transcription factor that plays an important role in the proliferation of tumor cells. Therefore, STAT3 is considered a target for anti-cancer therapy. In particular, IL-17 is involved in tumor growth promoted by STAT3 (27, 46, 47). However, the role of the IL-17/STAT3 pathway in promoting tumorigenesis upon activation by the oral microbiota from periodontitis is still unappreciated. Our TCGA database analysis showed that IL-17RA was highly expressed in HNSC. The correlation of IL-17A expression with the infiltration of γδ T cells and the expression of STAT3 was verified. The CCK-8 assay and immunofluorescence demonstrated the crucial role of IL-17 in STAT3 phosphorylation and SCC7 cell proliferation. A similar

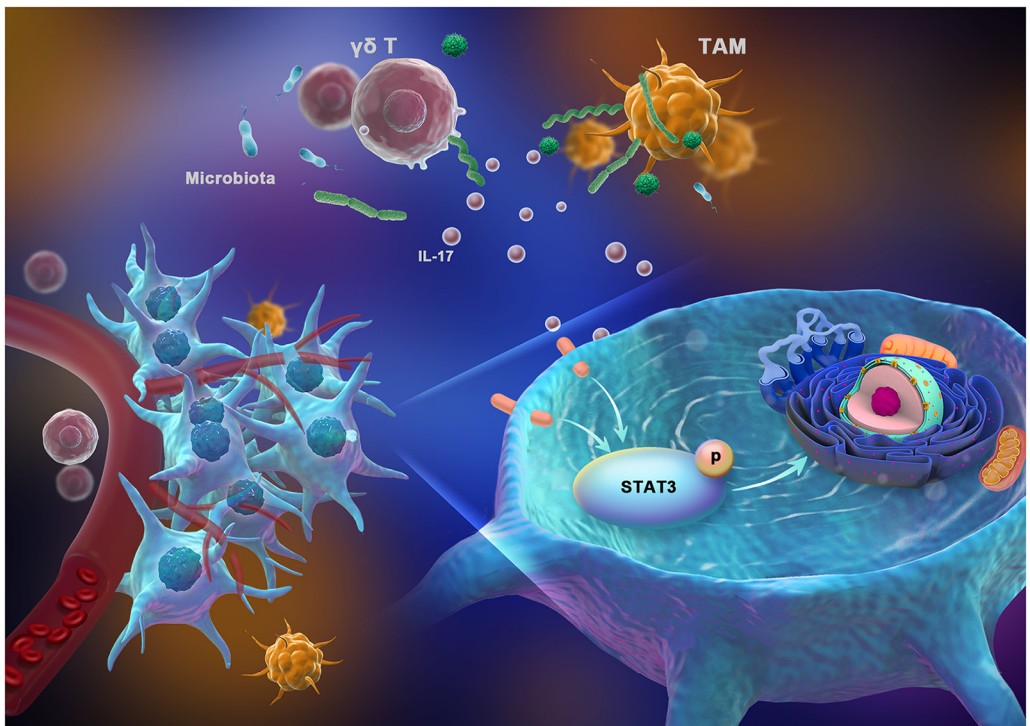

**FIG 7** Proposed mechanistic diagram. The tumor-associated immune response could be shaped by the oral microbiota from periodontitis. The oral microbiota from periodontitis drive the activation of IL-17$^+$ γδ T cells, which can promote tumor proliferation.

phenomenon in SCC7 cells could be observed in cells cocultured with PBMCs and the microbiota, which could be attributed to the high level of IL-17 concentration.

Our data indicated that IL-17$^+$ γδ T cells, IL-17A, and pSTAT3 had the same change trend *in vivo* and *in vitro*. Inhibition of γδ T cells led to decreases in the concentration of IL-17A, the phosphorylation level of STAT3, and the size of tumors. These new findings suggested that the IL-17/STAT3 pathway was regulated by γδ T cells in response to the oral microbiota from periodontitis, which were highly important to the development of tumors. Our *in vitro* data also showed that when the microbiota, cancer cells, and immune cells were present simultaneously, the abundance of M2-TAMs, the level of IL-17A secreted by γδ T cells, and the phosphorylation level of STAT3 in tumor cells peaked. Our study provided experimental solid evidence establishing cross talk among the microbiota, cancer cells, and the immune system. The tumor microenvironment is complex and controlled by multiple factors.

In summary, we identified an important pathway regulating OSCC immunity mediated by IL-17$^+$ γδ T cells in response to the oral microbiota from periodontitis. Our findings emphasize that the development and immune environment of OSCC are associated with alterations in the oral bacterial community composition. The oral microbiota from periodontitis drive the activation of IL-17$^+$ γδ T cells, and these IL-17$^+$ γδ T cells then promote tumor cell proliferation via the IL-17/STAT3 pathway (Fig. 7). Thus, oral commensal bacteria and IL-17$^+$ γδ T cells might be potential targets for monitoring and treating OSCC.

## MATERIALS AND METHODS

**Experimental mouse model.** Male BALB/c mice of 6 weeks of age were purchased from Dashuo Biological Technology (Chengdu, China). Mice were divided into groups randomly.

Before the bacterial inoculation, mice had drunk water with kanamycin (0.5 mg/mL; MCE catalog no. HY-16566A) for three consecutive days, and then the 5-0 silk ligatures were tied around the maxillary molars of mice. The microbiota inoculation came from the saliva of periodontitis patients or healthy people. We collected 2 mL saliva from each designated patient or healthy person. All the saliva was mixed, aliquoted, and cryopreserved quickly. After centrifugation of aliquoted saliva, the resulting pellet was

mixed with carboxymethyl cellulose and applied to the mouse oral cavity seven consecutive times. There was an interval of 1 day between each microbial inoculation operation. The P, OP, AOP, EOP, AOP-Anti, and AON groups were ligated and infected with oral microbiota from periodontitis patients or healthy people.

For the tumor inoculation, $5 \times 10^6$ mouse squamous carcinoma cells from cell line SCC7 in 50 $\mu$L Dulbecco modified Eagle medium (DMEM) were injected into the buccal mucosa of the mouse mouth. The survival rate curve recorded the number of surviving mice per day since SCC7 inoculation. OSCC tissues were analyzed on the 7th day or 21st day after injection. The O, OP, AO, AOP, AOA, EO, EOP, EOA, AON, AOP-Anti, and OSCC-with-ligature groups were treated with tumor inoculation.

For antibiotic treatment (20), ampicillin (1 g/L; Solarbio catalog no. A8180-1), neomycin trisulfate (1 g/L; MCE catalog no. HY-B0470), metronidazole (1 g/L; MCE catalog no. HY-B0318), and vancomycin (500 mg/L; MCE catalog no. HY-B0671) (4Abx) had been in the drinking water of mice continuously. The AOA and EOA groups were treated with 4Abx throughout the experimental period. The antibiotic drinking water treatment lasted for 29 days in the AOA and EOA groups (Fig. 1b).

For antibody or cytokine injection, mice were treated with the intraperitoneal injection of $\gamma\delta$-TCR monoclonal antibodies (200 $\mu$g/mouse; BioXCell catalog no. BE0070), once every 2 days.

**Human research participants.** The saliva from 4 designated chronic periodontitis patients was collected. The plaque was consistent with the degree of inflammation and destruction of periodontal tissue. The gingiva tissues were inflamed and bleeding on probing. The depth of the periodontal pocket was 4 to 6 mm, and X-ray films showed that the alveolar bone resorption exceeded one-third of the root length. The average age of the patients was 35 years. Four healthy people (average age, 27 years) were selected as the controls. All the volunteers did not have other maxillofacial or serious systemic infectious diseases and had not taken antibiotics, hormones, or antifungal drugs within 3 months. Exclusions also included surgery, radiation, chemotherapy, and pregnancy. OSCC patients were given periodontal examinations before surgery. The tumor tissues were collected intraoperatively. A total of 10 paraformaldehyde-fixed OSCC tissues were collected (OSCC without periodontitis, $n = 4$; OSCC with periodontitis, $n = 6$).

**H&E staining.** The tumors of mice were processed and embedded in paraffin or OCT compound (Sakura catalog no. 4583). Five-micrometer sections were prepared. Hematoxylin and eosin (H&E) staining was performed according to the manufacturer's instructions (Solarbio catalog no. G1120).

**Immunohistochemistry (IHC).** The Ki67 (Cell Signaling Technology catalog no. 12202) and TLR4 (Abcam catalog no. ab13556) primary antibodies were used to stain the sections. After treatments with trypsin, 0.5% phosphate-buffered saline with Tween 20 (PBST), hydrogen peroxide solution, and goat serum, the sections were incubated with Ki67 or TLR4 antibodies at 4°C. The sections were detected by the Universal SP kit (ZSGB-Bio catalog no. SP-9000). The images were analyzed and quantified by ImageJ (V1.53) software after being captured under the microscope (Leica Application Suite X software).

**Methylene blue staining.** The tissues around the teeth were removed. After bleaching the teeth and jaw with NaOCl and 3% $H_2O_2$, we used 1% methylene blue to stain the maxilla. The area of bone resorption, from the cementoenamel junction (CEJ) to the alveolar bone crest (ABC), was measured.

**16S rRNA sequencing.** Mouse oral saliva was collected with a sterile swab. DNA from each sample was extracted with the TIANamp bacterial DNA kit (Tiangen catalog no. DP302). The 16S rRNA gene was amplified with primers 5'-ACTCCTACGGGAGGCAGCA-3' and 5'-TCGGACTACHVGGGTWTCTAAT-3'. The sequencing areas were the V3 and V4 regions. After the construction of the library, sequence processing was performed using the MEM (Deng Lab) pipeline (48) (http://mem.rcees.ac.cn). UPARSE was used to classify the sequences into operational taxonomic units (OTUs). Resampling of 43,302 reads per sample was used to normalize. $\alpha$ diversity was calculated by Chao, richness, evenness, and phylogenetic diversity of the 97% identity OTUs. PCoA and CCA plots were made by the MEM (Deng Lab) pipeline. Linear discriminant analysis (LDA) effect size (LEfSe) analysis was run by the Genescloud pipeline (https://www.genescloud.cn), and metabolic pathway prediction was run by Phylogenetic Investigation of Communities by Reconstruction of Unobserved States (PICRUSt2, https://github.com/picrust/picrust2/wiki) and MetagenomeSeq (https://www.genescloud.cn).

**qRT-PCR.** Total RNA was extracted using an RNA extraction kit (BioTeke catalog no. RP1202). After reverse transcription (TaKaRa catalog no. RR036A), the cDNA samples were detected by a qRT-PCR kit (TaKaRa catalog no. RR820A). The mRNA expression levels of *Tlr4*, *Il10*, *Ifnγ*, *Il6*, *Il1β*, and *Il18* were normalized to that of *Gapdh*.

**ELISA.** IL-17 and IFN-$\gamma$ levels in the serum, the tumor tissues, and the cell culture supernatant were measured by ELISA detection according to IL-17 (BioLegend catalog no. 436204) and IFN-$\gamma$ (BioLegend catalog no. 430804) kit instructions.

**Flow cytometry.** The OSCC tissues were cut and digested into single cells. Dissociated cells (from the OSCC tissues or *in vitro* experiment) were filtered with a 70-$\mu$m cell strainer. After being stained with the Fixable Viability kit (BioLegend catalog no. L423105), single cell suspensions were blocked with anti-CD16/32 antibody (BioLegend catalog no. 101320) and incubated with the following antibodies: CD45 (BioLegend catalog no. 103140), CD3 (BioLegend catalog no. 100204), TCR $\gamma\delta$ (BioLegend catalog no. 118118; BioLegend catalog no. 107508), PD-1 (BioLegend catalog no. 135206; BioLegend catalog no. 114101), IL-17A (BioLegend catalog no. 506916), CD11b (BioLegend catalog no. 101228), F4/80 (BioLegend catalog no. 123130), CD206 (BioLegend catalog no. 141706). For intracellular cytokine staining, single cell suspensions were stimulated with cell activation cocktail (BioLegend catalog no. 423303) at 37°C in a 5% $CO_2$ incubator for 6 h before surface staining, fixation (BioLegend catalog no. 420801), and permeabilization (BioLegend catalog no. 421002). Flow cytometry was performed on an Attune NxT

flow cytometer (Invitrogen Attune NxT flow cytometry software), and data were analyzed by the FlowJo (V10.8) software.

**Immunofluorescence.** The samples were treated with trypsin, permeabilized by 0.5% PBST, peroxided by hydrogen peroxide solution, and blocked in goat serum. The following antibodies were used: anti-TCR $\gamma\delta$ (BioLegend catalog no. 118101; BioLegend catalog no. 331201), anti-CD206 (Proteintech catalog no. 18704-1-AP), anti-IL-17 (Proteintech catalog no. 26163-1-AP), anti-PD-1H (BioLegend catalog no. 143702), and anti-pSTAT3 (Cell Signaling Technology catalog no. 9145). The secondary antibodies (BioLegend catalog no. 405510; Abcam catalog no. ab6717; Abcam catalog no. ab150115) and 4',6-diamidino-2-phenylindole (DAPI) (Solarbio catalog no. C0065) were incubated successively. After staining, slides were examined on an Olympus confocal microscope (FV31S-SW V2.4 software).

***In vitro* experiment.** Peripheral blood lymphocyte separation solution (Solarbio catalog no. P6340, catalog no. P8900) was used to extract PBMCs from mouse or human blood. PBMCs were cultured in RPMI 1640. SCC7 cells or Cal27 cells (ATCC catalog no. CRL-2095) were cultured in DMEM. The oral microbiota came from the saliva of designated periodontitis patients. The saliva was mixed and centrifuged, and the resulting pellet was weighted. For ultrasonication, we added the lysate (50 mM Tris-HCl, 100 $\mu$g/mL lysozyme, 0.2 mM EDTA, 0.1% Triton X-100) to the pellet at a ratio of 1:10 and ultrasonically (300 W for 20 min) broke it on ice. For heat inactivation, the pellet was sterilized at 70°C for 10 min and dispersed by high-pressure homogenization (49). The *Porphyromonas. gingivalis* strain W83 was grown under anaerobic conditions at 37°C in brain heart infusion broth containing 5 $\mu$g/mL hemin and 0.5 $\mu$g/mL menadione. The CFU per milliliter was measured.

PBMCs ($5 \times 10^5$) were cocultured with 1.5 $\mu$g live oral microbiota from periodontitis for 1 or 6 h. With or without filters, $5 \times 10^5$ PBMCs were cocultured with 1.5 $\mu$g microbiota treated in different ways for 6 h, respectively. PBMCs ($5 \times 10^5$) were infected with *P. gingivalis* (multiplicity of infection [MOI] = 100) for 6 h. With or without filters, $5 \times 10^6$ SCC7 cells or Cal27 cells (ATCC catalog no. CRL-2095) were cocultured with the live microbiota (1.5 $\mu$g), PBMCs ($5 \times 10^5$ cells) from mouse or human, *P. gingivalis* (MOI = 100), and 200 ng/mL IL-17 for 6 h, respectively, in 12-well plates (Corning catalog no. 3401). With or without filters, $5 \times 10^3$ SCC7 cells were cocultured with 200 ng/mL IL-17, the live oral microbiota from periodontitis (1.5 ng), and PBMCs ($5 \times 10^2$ cells), respectively, in 96-well plates (Corning catalog no. 3381). PBMCs were analyzed by flow cytometry. The cell culture supernatant was measured by ELISA. The SCC7 cells or the Cal27 cells were analyzed by immunofluorescence or CCK-8 assay.

**Cell counting kit 8.** The CCK-8 kit (Biosharp catalog no. BS350B) was used to measure the cell vitality of SCC7 cells. After washing with PBS, CCK-8 was added after 6, 12, 24, and 48 h. The absorbance value of each well was measured at 450 nm.

**WB.** Total proteins were extracted using a protein extraction kit (SAB catalog no. PE001). The Western blot (WB) experiment was performed according to the standard procedures. Primary antibodies against phosphorylated STAT3 (Cell Signaling Technology catalog no. 9145), STAT3 (Cell Signaling Technology catalog no. 9139), and glyceraldehyde-3-phosphate dehydrogenase (GAPDH) (Proteintech catalog no. 60004-1-lg) were used at a 1:1,000 dilution.

**Clinical data analysis.** The gene expression levels of IL-17RA in HNSC tissues and normal control tissues were compared within the TCGA database (https://cancergenome.nih.gov/). The significance was assessed by Mann-Whitney U test. According to the TIMER software (https://cistrome.shinyapps.io/timer/), the correlations of IL-17A and $\gamma\delta$ T cell infiltration level in HNSC were searched. The correlation analysis of IL-17A or IL-17RA and STAT3 gene expression was calculated within the TCGA database.

**Study approval.** This animal study was approved by the Ethics Committee of West China School of Stomatology, Sichuan University (protocol number: WCHSIRB-D-2019-015). All experiments were approved and carried out according to the Guide for the Care and Use of Laboratory Animal (50). For humans, all volunteers were from the West China Hospital of Stomatology, Sichuan University. The study was carried out under the approval and supervision of the Medical Ethics Committee of West China Hospital of Stomatology, Sichuan University (WCHSIRB-OT-2019-015), and conducted in accordance with the Declaration of Helsinki. The written informed consents were signed for each participant.

**Statistical analysis.** Data are represented as the mean $\pm$ standard deviation (SD) for independent samples. Analysis of variance (ANOVA) for parametric data and Mann-Whitney U test for nonparametric data were applied for data analysis. Student's $t$ test was used for comparing two groups. The $\alpha$ diversity was analyzed by Kruskal-Wallis test and Dunn's test. The significance of LEfSe was determined by an LDA score of >2.0 and a $P$ value of <0.05 for the Kruskal-Wallis test. The correlation coefficient was analyzed by Shapiro-Wilk normality test and Pearson correlation coefficient calculation. $P$ values of <0.05 or U values of >1.96 were considered significant statistically. All poststudy statistical analysis was performed using GraphPad Prism v.7.04.

**Data availability.** All data generated or analyzed during this study are included in this published article. The 16S rRNA sequencing data (PRJCA006656 or CRA005049) have been deposited in the Genome Sequence Archive (Genomics, Proteomics & Bioinformatics 2021), accessible at https://ngdc.cncb.ac.cn/gsa.

## SUPPLEMENTAL MATERIAL

Supplemental material is available online only.

**FIG S1**, TIF file, 2 MB.
**FIG S2**, TIF file, 2.4 MB.
**FIG S3**, TIF file, 2.5 MB.

**FIG S4**, TIF file, 0.9 MB.
**FIG S5**, TIF file, 2.5 MB.

## ACKNOWLEDGMENTS

This study was supported by the National Natural Science Foundation of China (81771085, 81991500, 81991502, 81970944), Key Projects of Sichuan Provincial Department of Science and Technology (2020YFSY0008).

We thank Hong Li and Jiao Cheng for valuable advice on flow cytometry analysis and data analysis.

Y. Li conceived the study. W. Wei, J. Li, J. Lyu, X. Shen, C. Yan, W. Ma, and B. Tang performed laboratory assays and experiments, supervised by Y. Li. W. Wei, J. Li, J. Lyu, X. Shen, C. Yau, B. Tang, and Y. Li analyzed the laboratory data. W. Wei and Y. Li produced the tables and figures. W. Wei and Y. Li wrote the first draft with assistance from H. Xie, L. Zhao, L. Cheng, and Y. Deng. All authors critically reviewed and approved the final manuscript.

We declare that no conflict of interest exists.

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
