## [Reviewer comments · mSystems]

Oral microbiota from periodontitis promote oral squamous cell carcinoma development via $\gamma\delta$ T activation

Wei Wei, Jia Li, Xin Shen, Jinglu Lyu, Caixia Yan, Boyu Tang, Wenjuan Ma, Huixu Xie, Lei Zhao, Lei Cheng, Ye Deng, and Yan Li

Corresponding Author(s): Yan Li, Sichuan University

Review Timeline:

Submission Date:	May 18, 2022
Editorial Decision:	July 10, 2022
Revision Received:	August 3, 2022
Accepted:	August 3, 2022

Editor: Jack Gilbert

Reviewer(s): Disclosure of reviewer identity is with reference to reviewer comments included in decision letter(s). The following individuals involved in review of your submission have agreed to reveal their identity: Vivek Thumbigere-Math (Reviewer #1)

Transaction Report:

DOI: <https://doi.org/10.1128/msystems.00469-22>

July 10, 2022

Dr. Yan Li
Sichuan University
Chengdu
China

Re: mSystems00469-22 (Oral microbiota from periodontitis promote oral squamous cell carcinoma development via $\gamma\delta$ T activation)

Dear Dr. Yan Li:

Thank you for submitting your manuscript to mSystems. We have completed our review and I am pleased to inform you that, in principle, we expect to accept it for publication in mSystems. However, acceptance will not be final until you have adequately addressed the reviewer comments.

Preparing Revision Guidelines

Sincerely,

Jack Gilbert

Editor, mSystems

Journals Department
American Society for Microbiology
1752 N St., NW

Reviewer comments:

Reviewer #1 (Comments for the Author):

This is a translation study investigating the role of periodontitis and associated microbiota in the promotion of OSCC development. The study shows oral microbiota from periodontitis patients promote RorGT cell mediated IL17-STAT3 pathway and tumor associated M2 macrophages as a fundamental pathway in OSCC development. While this study is significant, there are few weaknesses that need to be addressed -

1. The author state that ligation in mice did not affect tumor size development and proliferation without the inoculation of microbes from periodontitis patients. If so, is human oral bacteria more critical than native mice oral microbiota in OSCC development.
2. It's unclear if saliva from aggressive or chronic periodontitis patients were inoculated in mice?
3. It appears that advanced OSCC with Antibiotic (AOA) had the highest tumor burden and the authors attribute this to continuous use of antibiotic that might have resulted in the formation of drug-resistant bacterial community. The animals were treated with 3 days of antibiotics before oral bacterial inoculation. Three days of treatment will not lead to drug resistance.
4. The study shows that periodontitis related microbiota plays a major role in OSCC development. However, it is unclear which are the major pathogens involved. It's important to look into specific effects of *P. gingivalis* at least. Need to perform in vitro studies using *P. gingivalis*.
5. Following anti-RorGT treatment, was there changes in oral microbiota that may have led to reduced tumor burden?

Point-by-point responses to the reviewers' comments and criticisms:

Editor's Remarks to Author:

Thank you for submitting your manuscript to mSystems. We have completed our review and I am pleased to inform you that, in principle, we expect to accept it for publication in mSystems.

However, acceptance will not be final until you have adequately addressed the reviewers' comments.

Response: We thank the Editor for giving us this valuable opportunity to resubmit our current revised manuscript. We also addressed the Reviewers' comments by preparing point-by-point responses below.

Preparing Revision Guidelines

To submit your modified manuscript, log onto the eJP submission site at

<https://msystems.msubmit.net/cgi-bin/main.plex>. Go to Author Tasks and click the appropriate

manuscript title to begin the revision process. The information that you entered when you first submitted the paper will be displayed. Please update the information as necessary.

Response: We appreciate the efforts of the Editor in bringing this important issue to our attention.

- Point-by-point responses to the issues raised by the reviewers in a file named "Response to Reviewers," NOT IN YOUR COVER LETTER.

Response: We appreciate the Editor's reminder. We have prepared a separate letter entitled "Response to Reviewers" according to the requirements.

- Upload a compare copy of the manuscript (without figures) as a "Marked-Up Manuscript" file.

Response: We appreciate the Editor's reminder. We have prepared a comparison copy of the manuscript entitled "Marked-Up Manuscript" according to the requirements. All the modified text is highlighted in the revised manuscript.

- Each figure must be uploaded as a separate file, and any multipanel figures must be assembled into one file.

Response: We appreciate the editor's reminder. We have checked all the figures and made sure that they could meet the requirements.

- Manuscript: A .DOC version of the revised manuscript

Response: We appreciate the Editor's reminder. We have prepared a .DOC version of the revised manuscript

- Figures: Editable, high-resolution, individual figure files are required at revision, TIFF or EPS files are preferred

Response: We appreciate the editor's reminder. We have checked the quality of all figures and made sure that they could meet the requirements.

Response: We appreciate the editor's reminder. We have verified all links to sequence records and made sure that each number retrieves the full record of the data.

Reviewer comments:

Reviewer #1 (Comments for the Author):

This is a translation study investigating the role of periodontitis and associated microbiota in the promotion of OSCC development. The study shows oral microbiota from periodontitis patients promote RorgT cell mediated IL17-STAT3 pathway and tumor associated M2 macrophages as a fundamental pathway in OSCC development. While this study is significant, there are few weaknesses that need to be addressed.

Response: We would like to thank the Reviewer for considering that our study was significant.

We have addressed the Reviewer's concerns by revising our manuscript and have responded to the comments point-by-point below.

1. The author state that ligation in mice did not affect tumor size development and proliferation without the inoculation of microbes from periodontitis patients. If so, is human oral bacteria more critical than native mice oral microbiota in OSCC development.

Response: We appreciate the efforts of the reviewer in bringing this important issue to our attention. In our study, we performed a variety of interventions on mice, such as ligation alone, ligation with inoculation of healthy human oral bacteria (AON group), ligation with inoculation of periodontitis patient oral bacteria (OP/EOP/AOP group), and long-term antibiotic treatment (EOA/AOA group). Our experiment indicated that the changes in the mouse native oral microbiota and mouse immunity caused by ligation alone did not significantly contribute to the development of OSCC (Supplementary Fig. 1c). In addition, we found that the healthy human oral bacteria also did not significantly affect the development of OSCC, but periodontitis oral bacteria did (Fig. 5). At the same time, 16S rRNA sequencing showed that the periodontitis oral bacteria dramatically altered the mice oral commensal bacterial community in OSCC, and some particular oral microbes from periodontitis could be the dominant "king" in the entire process of tumor development (Fig. 2). After comparison, we thought that although the human oral bacteria invaded the native oral ecology of mice, certain pathogenic oral bacteria of periodontitis did play a key role in promoting OSCC. Some studies showed that the key human periodontitis oral bacteria, such as *Porphyromonas* and *Fusobacterium*, were associated with the development of

human OSCC (Chen, Q. et al. Salivary Porphyromonas gingivalis predicts outcome in oral squamous cell carcinomas: a cohort study. BMC Oral Health. 2021. 21, 228; Juliana D Bronzato. et al. Detection of Fusobacterium in oral and head and neck cancer samples: A systematic review and meta-analysis. Archives of Oral Biology. 2020. 112, 104669), which supports our conclusion.

In conclusion, we believe that the human periodontitis oral bacteria need more attention in the study of oral squamous cell carcinoma. As suggested, we added the discussion, and please refer to the revised manuscript (Pages 20-21, Lines 281-289).

2. It's unclear if saliva from aggressive or chronic periodontitis patients were inoculated in mice?

Response: We thank the reviewer for pointing this out, and we apologize for the missing information. The periodontitis saliva was from designated chronic periodontitis patients. The volunteers with periodontitis were all adults, and their teeth had obvious plaque, which was consistent with the degree of inflammation and destruction of periodontal tissue. The gingiva tissues were inflamed and bleeding on probing. The depth of the periodontal pocket was 4-6 mm, and X-ray films showed that the alveolar bone resorption exceeded 1/3 of the root length. As

suggested, we have added detailed information in the Materials and Methods section. Please refer to the revised manuscript (Page 27, Lines 376-380).

3. It appears that advanced OSCC with Antibiotic (AOA) had the highest tumor burden and the authors attribute this to continuous use of antibiotic that might have resulted in the formation of drug-resistant bacterial community. The animals were treated with 3 days of antibiotics before oral bacterial inoculation. Three days of treatment will not lead to drug resistance.

Response: We thank the reviewer for pointing this out, and we apologize for the missing information. The AOA and EOA groups were treated with 4Abx throughout the experimental period. The antibiotic drinking water treatment lasted for 29 days in the AOA and EOA groups (Fig. 1b). 16S rRNA sequencing showed that the α diversity of the oral microbiome decreased under the 4Abx treatment (EOA and AOA groups) (Fig. 2a). As suggested, we have added detailed information in the Materials and Methods section. Please refer to the revised manuscript (Page 26, Lines 370-371).

4. The study shows that periodontitis related microbiota plays a major role in OSCC development.

However, it is unclear which are the major pathogens involved. It's important to look into specific effects of *P. gingivalis* at least. Need to perform in vitro studies using *P. gingivalis*.

Response: We appreciate the reviewer's insightful commentary and valuable suggestions. Our

16S rRNA sequencing showed that *Porphyromonas* was the most abundant genus in the EOP

(92.46%) and AOP groups (47.58%) (Fig. 2c). *Fusobacterium*, *Neisseria*, and *Leptotrichia* were

more abundant in the AOP group (Supplementary Fig. 2b). As suggested, we performed *in vitro*

experiments to look into the specific effects of *P. gingivalis*. Although the proportion of $\gamma\delta$ T

cells in PBMCs did not increase after co-culturing with *P. gingivalis*, IL-17⁺ $\gamma\delta$ T cells

proliferated more than those in the group without *P. gingivalis* (Supplementary Fig. 4a). At the

same time, a significant increase in IL-17A was observed (Supplementary Fig. 4b). When

PBMCs and *P. gingivalis* were co-cultured with cancer cells, the increased expression of

pSTAT3 in these cancer cells was observed (Supplementary Fig. 4c). O Barel et al. found the

higher expression levels of IL-17 in the *P. gingivalis* infection mice, and $\gamma\delta$ T might play an

important role (O Barel et al. $\gamma\delta$ T Cells Differentially Regulate Bone Loss in Periodontitis

Models. J Dent Res. 2022. 101(4):428-436). The data of O Barel et al. could help to prove the results of our *in vitro* experiments. We have added the detailed information in our revised paper.

Please refer to Page 15, Lines 198-202; Page 18, Lines 246-248; Pages 32-33, Lines 460-469.

5. Following anti-Rorg treatment, was there changes in oral microbiota that may have led to reduced tumor burden?

Response: We appreciate the reviewer bringing our attention to this crucial issue. Although the purpose of this study is to find out how oral microbiota from periodontitis promote oral squamous cell carcinoma development via $\gamma\delta$ T, the crosstalk between $\gamma\delta$ T cells and the microbiota has been underappreciated. Shi et al. discovered a close positive correlation between $\gamma\delta$ T cells and the α -diversity of the microbiota in the lungs of cancer patients (Shi, et al. Lung microbiota: Unexploited treasure hidden in the immune microenvironment of lung cancer.

Thoracic Cancer. 2021. 12:2964-2966). Wilharm et al. found that ablation of $\gamma\delta$ T cells alters the relative diversity of oral microbiota in SPF B6 mice. (Wilharm, et al. Mutual interplay between IL-17–producing $\gamma\delta$ T cells and microbiota orchestrates oral mucosal homeostasis. PNAS. 2019.

116 (7): 2652-2661). We are constantly monitoring the microbe. Our 16S rRNA sequencing showed that the α and β diversity of oral microbiota did not change significantly after inhibition of $\gamma\delta$ T, but we will further analyze the oral microbiota by Metagenome and Metabolome Sequencing in the future. However, studies of the interplay between the resident microbiota and $\gamma\delta$ T cells are limited, and additional mechanistic insights remain to be exploited in the future. In the Discussion section, we have added the latest research reports on the crosstalk between $\gamma\delta$ T cells and microbiota. Please refer to the revised manuscript (Page 22, Lines 308-313) for details.

August 3, 2022

Dr. Yan Li
Sichuan University
Chengdu
China

Re: mSystems00469-22R1 (Oral microbiota from periodontitis promote oral squamous cell carcinoma development via $\gamma\delta$ T activation)

Dear Dr. Yan Li:

Your manuscript has been accepted, and I am forwarding it to the ASM Journals Department for publication. For your reference, ASM Journals' address is given below. Before it can be scheduled for publication, your manuscript will be checked by the mSystems production staff to make sure that all elements meet the technical requirements for publication. They will contact you if anything needs to be revised before copyediting and production can begin. Otherwise, you will be notified when your proofs are ready to be viewed.

Publication Fees:

If you would like to submit a potential Featured Image, please email a file and a short legend to mSystems@asmusa.org. Please note that we can only consider images that (i) the authors created or own and (ii) have not been previously published. By submitting, you agree that the image can be used under the same terms as the published article. File requirements: square dimensions (4" x 4"), 300 dpi resolution, RGB colorspace, TIF file format.

We recognize that the video files can become quite large, and so to avoid quality loss ASM suggests sending the video file via <https://www.wetransfer.com/>. When you have a final version of the video and the still ready to share, please send it to mSystems staff at mSystems@asmusa.org.

Sincerely,

Jack Gilbert
Editor, mSystems

Journals Department
Fig. S1: Accept

Fig. S2: Accept

Fig. S3: Accept

Fig. S5: Accept

Fig. S4: Accept